# The Burden of Hidradenitis Suppurativa Signs and Symptoms in Quality of Life: Systematic Review and Meta-Analysis

**DOI:** 10.3390/ijerph18136709

**Published:** 2021-06-22

**Authors:** Trinidad Montero-Vilchez, Pablo Diaz-Calvillo, Juan-Angel Rodriguez-Pozo, Carlos Cuenca-Barrales, Antonio Martinez-Lopez, Salvador Arias-Santiago, Alejandro Molina-Leyva

**Affiliations:** 1Dermatology Department, Hospital Universitario Virgen de las Nieves, Avenida de Madrid, 15, 18012 Granada, Spain; tmonterov@correo.ugr.es (T.M.-V.); pdc.muro@gmail.com (P.D.-C.); juanangelrpg@gmail.com (J.-A.R.-P.); carloscuenca1991@gmail.com (C.C.-B.); antoniomartinezlopez@aol.com (A.M.-L.); alejandromolinaleyva@gmail.com (A.M.-L.); 2Instituto de Investigación Biosanitaria GRANADA, 18012 Granada, Spain; 3Dermatology Department, Faculty of Medicine, University of Granada, 18001 Granada, Spain

**Keywords:** acne inversa, dermatology, hidradenitis suppurativa, quality of life, pain, pruritus

## Abstract

Hidradenitis suppurativa (HS) is a chronic, recurrent and debilitating inflammatory skin disease of the hair follicle that usually presents as painful, deep-seated inflamed lesions in the apocrine gland-bearing areas of the body. HS patients suffer from uncomfortable signs and symptoms, such as pain, pruritus, malodour and suppuration, which may impair patients’ quality of life (QoL). Although HS patients frequently experience these signs and symptoms, they are only occasionally assessed by clinicians and, unexpectedly, the scientific evidence available is limited and heterogeneous. The aim of this study is to summarize the evidence regarding the impact of HS signs and symptoms on QoL to serve as a basis for future research and help clinicians to consider them in the daily care of HS patients. A systematic review and meta-analysis were conducted following PRISMA Guidelines. The following search algorithm was used: (hidradenitis or “acne inversa”) and (pain or itch or odour or malodour or suppuration or oozing or drainage) and (“quality of life”). The literature search identified 836 references, 17 of them met the eligible criteria and were included for analysis, representing 4929 HS patients. Mean age of the participants was 36.28 years and there was a predominance of female sex among study participants. The BMI of the population was in the range of over-weight and about two out five patients were active smokers. Studies included patients with mild to moderate HS, with a mean disease duration of 13.69 years. The HS signs and symptoms assessed were pain, pruritus, malodour and suppuration. Overall, the higher intensity of a sign or symptom correlated with poorer general QoL or specific QoL dimensions including sexual distress, anxiety, depression and sleep. The most frequently employed tool to assess QoL was the Dermatology Life Quality Index (DLQI). DLQI was used in 52.9% of the studies (9/17) with a mean value of 10.70 (2.16 SD). The scores employed to assess signs and symptoms severity were subjective and varied between studies, being the numerical rating scale (NRS) for each of the most used symptoms. The mean NRS value for pain was 3.99 and the mean NRS for pruritus was 4.99. In conclusion, we have summarized, categorized and analyzed the scientific evidence regarding signs and symptoms in HS patients and their impairment in QoL. Their assessment should be thorough and included during routine evaluation of HS patients to motivate therapeutic modifications and increase patients’ health.

## 1. Introduction

Hidradenitis suppurativa (HS) is a chronic, recurrent and debilitating inflammatory skin disease of the hair follicle that usually presents after puberty with painful, deep-seated inflamed lesions in the apocrine gland-bearing areas of the body, most commonly the axillae, inguinal and anogenital regions [1,2]. It has an estimated prevalence rate in the general population of 1% [3] and is associated with several comorbidities, such as metabolic syndrome, cardiovascular risk, diabetes or inflammatory bowel disease [4].

HS is one of the dermatological diseases with the greatest impact on patients’ quality of life (QoL) [5]. Its impairment is similar to other conditions, such as cardiovascular disease, cancer, diabetes mellitus and chronic obstructive pulmonary disease [6]. In fact, the mean Dermatology Life Quality Index scores for HS (8.3–12.7) are typical for severe dermatoses [7]. The disease severity, the number of flares and the lesion location are major factors impairing QoL [7,8]. HS not only impairs the physical health but also mental and psychosocial health. Poor self-esteem and body image [9,10] and increased risk of anxiety and depression [11,12] are also factors associated with HS that worsen patients’ life. Moreover, HS has a negative effect on sexual function [13].

The tool used most frequently to assess patients’ QoL is the DLQI. Recently, disease specific instruments to assess quality of life in HS have been developed [14], such as the HSQoL-24 validated in the Spanish population [15]. Other questionnaires also employed to evaluate psychosocial and physical functioning in HS are Skindex, the EuroQol 5 Dimensions questionnaire (EQ-5D) and the Short Form 36 questionnaire (SF-36). The Hospital Anxiety and Depression Scale (HADS), Beck Depression Inventory (BDI) and Major Depression Inventory (MDI) have been used to assess the impact on psychological QoL [7]. The influence of sexual function on patients’ life has also been evaluated using different questionnaires, such as the six-item Female Sexual Function Index (FSFI-6) and the Five-Item International Index of Erectile Function (IIEF-5) [13].

The great impact on QoL might be in part due to its uncomfortable signs and symptoms. HS lesions produce pain, pruritus, malodour and suppuration, which make life difficult for patients [16,17]. Pain seems the most common and bothersome symptom of HS and it is usually linked to the inflammatory nodules or abscesses, reported by more than 95% of patients [7]. The pain reported in HS patients is higher than other skin diseases [18]. The pain is not specific to any location on the body and it is mainly described as shooting, itchy and blinding [19]. Regarding other HS symptoms, pruritus is the other one most overlooked in the literature, although HS is not usually considered a pruritic disease [20]. The most common tools used to evaluate the severity of HS symptoms are the visual analogue scale (VAS) and the numerical rating scale (NRS) [7].

Pain, pruritus, malodour and suppuration are signs and symptoms frequently experienced by patients, but they are only occasionally assessed by clinicians [21]. Signs and symptoms might be the main burden of patients with HS, producing a great impairment in quality of life. Unexpectedly, the scientific evidence available is limited and heterogeneous. The aim of this study is to summarize the evidence regarding the impact of HS signs and symptoms on QoL to serve as a basis for future research and help clinicians to consider them in the daily care of HS patients.

## 2. Materials and Methods

A systematic review and meta-analysis were conducted. A literature search was performed using Medline, Scopus and Embase databases from conception to 4 May 2021, following PRISMA Guidelines (Appendix A). The following search algorithm was used: (hidradenitis or “acne inversa”) and (pain or itch or odour or malodour or suppuration or oozing or drainage) and (“quality of life”). Symptoms included in the literature search were selected by a dermatologist expert in HS (AML) following the most recent evidence in HS clinical presentation [1].

The search was limited to: (i) human data, (ii) articles correlating HS symptoms with quality-of-life impairment in HS patients, (iii) articles written in English. All types of epidemiological studies (clinical trials, cohort studies, case-control studies and cross-sectional studies) were included and analyzed. Reviews, guidelines, protocols, case series, case reports and conference abstracts were excluded.

Two researchers (TMV and AML) independently reviewed the titles and abstracts of the articles obtained in the first search to assess relevant studies. The full texts of all articles meeting the inclusion criteria were reviewed, and their bibliographic references were checked for additional sources. The articles considered relevant by both researchers were included in the analysis. Disagreements about inclusion or exclusion of articles were subjected to discussion until a consensus was reached. If not reached, resolution was achieved by discussion with a third researcher (SAS).

The variables assessed were study design, author, country, level of scientific evidence according to the Centre for Evidence-Based Medicine, number of participants, age, sex, BMI (kg/m^2^), smoking habit, disease duration, disease severity (Hurley stage), HS symptoms and aspects of QoL evaluated, QoL and symptoms assessment tools and scores, correlation between symptoms and QoL.

The mean DLQI and NRS for symptoms was calculated by a random effect meta-analysis weighted by the study sample size. To estimate absolute mean effect of DLQI and NRS for each symptom, the mean, standard deviation and sample size were extracted from the studies. Research with unclear or incomplete reporting was excluded from the meta-analysis. To generate valid estimates, studies were weighed according to their sample size. Forest plots were constructed to assess the distribution of the data and summarize the effect size and their 95% CIs. Quantifying of Heterogeneity was evaluated using Cochrane Q statistic, an intermediary statistic employed to obtain a more useful measure of heterogeneity, the I2. Assuming a high heterogenicity between studies, we used a random effects model to calculate the outcome. Microsoft Excel version 2016, Redmond, Washington, The USA, was used to run this data [22].

The quality of the design was critically appraised using the National Institutes of Health quality assessment tool to evaluate risk of bias (Appendix A) [23]. This tool is based on the key concepts for evaluating the internal validity of a study and is divided into a set of 14 set questions. Studies are classified depending on the rate: good quality (>9 criteria met), fair quality (5–9 criteria met) and poor quality (<5 criteria met).

## 3. Results

The literature search identified 836 references, 523 after removing duplicated papers. After reviewing the title and abstract, 92 records underwent full-text review. A total of 75 records were excluded because they did not investigate the impact of HS symptoms on QoL. Other reasons for exclusion along with the flow chart are shown in Figure 1. Finally, 17 studies, representing 4929 patients with HS, met the eligible criteria and were included and fully reviewed [24,25,26,27,28,29,30,31,32,33,34,35,36,37,38,39,40].

The main characteristics of the studies included are summarized in Table 1. All studies had a cross-sectional design and were classified as scientific level of evidence 4. Samples were recruited from outpatient clinics or through focused electronic, postal or telephone surveys. Study participants were predominantly female. Mean age of the participants was 36.28 years. The BMI of the population was in the range of overweight, about two out five patients were active smokers. Studies included patients with mild to moderate HS, with a mean disease duration of 13.69 years. The body regions more frequently affected by HS were axilla and groins.

The HS signs and symptoms assessed were pain, pruritus, malodour and suppuration. Overall, the higher intensity of a sign or symptom correlated with poorer general QoL or specific QoL dimensions including sexual distress, anxiety, depression and sleep. The most frequently employed tool to assess QoL was the DLQI. DLQI was used in 52.9% of the studies (9/17) with a mean value of 10.70 (2.16 SD) after conducting a random effect meta-analysis weighted by the study sample size (Figure 2).

### 3.1. Pain

Thirteen studies evaluated the impact of pain in the QoL of HS patients, including 4216 participants with a mean age of 35.62 years (Table 2) [24,25,26,27,28,29,30,31,32,33,34,35,36].

The incidence of pain was reported in two studies, ranging from 65.24% [33] to 77.5% [32]. NRS was the scale most used to assess pain (53.85%, 7/13) [24,25,30,31,32,33,34]. The mean NRS value was 3.99 (SD 0.95) after conducting a random effect meta-analysis weighted by the study sample size (Figure 3). VAS was the second most employed tool for pain but was scored in different ways [26,27,29,32,35]. PainDETECT [27], boil-associated pain score [28] and self-reported number of painful lesions [36] were other ways of pain severity assessment.

The most frequently employed tool to assess QoL was the DLQI [26,27,28,29,31,32,33,35]. Skindex-29 [26], patient global assessment (PtGA) of QoL [30] and Skindex-17 [35] were also employed to assess overall QoL in HS patients. The Hospital Anxiety and Depression Scale (HADs)-Anxiety was used to assess anxiety [26] and HADS-depression [26], Beck’s Depression Inventory [27] and Major Depression Inventory (MID) [34] were employed to evaluate depression. Sexual distress was assessed by NRS for HS impact on sex life, six-item Female Sexual Function Index (FSFI-6) and five-item International Index of Erectile Function (IIEF-5) [24,25]. The impairment of sleep was assessed by using the Athens Insomnia Scale (AIS) and the Pittsburgh Sleep Quality Index (PSQI) [29].

Pain was related to the overall impact on QoL, assessed by DLQI independently of the tool used for assessing QoL and pain severity [26,27,28,30,32,33,35,36]. High NRS for pain was associated with high DLQI [31,32,33,34] (r = 0.581, *p* < 0.001 [31]; r = 0.48, *p* < 0.001 [32]; β = 0.91 ± 0.12, R2 = 0.36, *p* < 0.001 [33]; r = 0.60, *p* < 0.05 [34]) and PtGA of QoL (r = 0.66, 0.6–0.71 95% CI) [30]. High VAS for pain values were also linked to higher DLQI (r = 0.457, *p* < 0.001 [26]; r = 0.48, *p* < 0.001 [32]), Skindex-29 [26] and Skindex-17 [35]. Matusiak et al. showed that the presence of pain was a more important factor for worsening QoL (*p* = 0.002) than disease severity (*p* = 0.04). They also observed that pain severity was related to increased sweating, heat and physical activity [32].

Two studies showed that HS pain worsened psychological QoL [27,34]. Nevertheless, Frings et al. showed that pain impairs patients’ anxiety (r = 0.304, *p* = 0.009) but not depression (r = 0.193, *p* = 0.105) [26]. Kaaz et al. also found that HS pain was related to poor sleep quality [29]. Moreover, it was observed that NRS for pain was related to impact on sex life (β = 0.15, *p* = 0.049) [25] and was a risk factor for sexual dysfunction in females (β = 0.1, *p* < 0.05) [24].

### 3.2. Pruritus

Eight studies evaluated the impact of pruritus on HS patients’ QoL, including 2059 participants with a mean age of 38.88 years (Table 3) [24,25,29,32,33,34,39,40].

The incidence of pruritus was reported in three studies (41.7% [32] vs. 57.3% [40] vs. 61.8% [33]). The NRS for pruritus was the scale most used to assess pruritus severity [24,25,32,33,34,39,40]. The mean NRS was 4.99 (0.96 SD), after conducting a random effect meta-analysis weighted by the study sample size, Figure 4. VAS for pruritus [29], 4-item itch questionnaire [32] and 5-D itch scale [40].

Matusiak et al. observed that the presence of pruritus did not have an impact on QoL [32], while Molina-Leyva et al. observed that the presence of NRS for pruritus > 3 was related with higher rates in DLQI score (β = 0.42 ± 0.11, R2 = 0.20, *p* < 0.001) [33]. Moreover, higher rates in VAS and NRS for pruritus were positively correlated with DLQI [32,34]. The impact of pruritus in overall QoL was also reflected by Riis et al. showing that higher NRS for pruritus were related to lower values in the EQ-5D (β = −0.017, *p* < 0.05) [39].

It was found that HS pruritus impaired sleep quality [29,40] and it was linked to poor mental health assessed by MDI [34]. Nevertheless, it was observed that NRS for pruritus did not have an impact on sex life (β = 0.03, *p* = 0.615) [25], neither in men nor in women [24].

Factors associated with increased risk of pruritus were Hurley III, higher number of regions affected, the female sex, being an active smoker, the intensity of suppuration and pain, having Crohn’s disease and not using statins [32,33,40].

### 3.3. Malodour

Six studies evaluated the impact of pain in the QoL of HS patients, including 1507 participants with a mean age of 38.59 years [24,25,33,37,38,39], Table 4.

The incidence of malodour was reported in two studies, rating from 88.24% [37] to 40.8% [33]. NRS was the most common scale used to assess malodour severity [24,25,33,37,39], ranging from 3.28 ± 3.58 [33] to 5.6 ± 3.38 [25]. HODS-odour was also used [38].

Malodour was related to poor QoL, assessed by DLQI (R2 = 0.17, F = 2.63, *p* = 0.064 [37]; β = 0.44 ± 0.11, R2 = 0.23, *p* < 0.001 [33]), the Skindex-19 (R2 = 0.39, F = 8.11, *p* < 0.001) [37], the Skindex-29 [38] and the EQ-5D [39]. Moreover, it was observed that NRS for malodour had an impact on sex life (β = 0.13, *p* = 0.035) [25] and it was a risk factor for sexual dysfunction in females (β = 0.07, *p* < 0.05) but not in men [24].

Factors associated with increased risk of pruritus were higher BMI, longer disease duration, high number of regions affected and the location on groin, upper thighs, and buttocks, high Hurley stage and intensity of suppuration [33,37].

### 3.4. Suppuration

Three studies evaluated the impact of pain in the QoL of HS patients, including 802 participants with a mean age of 38.83 years [24,25,38], Table 5.

NRS for malodour (6.48 ± 3.18) [24,25] and HODS-drainage [38] were the scales used to assess suppuration severity. It was found that suppuration was related to the overall impact on QoL, assessed by Skindex-29 (r = 0.614, *p* < 0.05) and HS-QoL overall (r = 0.745, *p* < 0.05^) [38]. HODS-drainage was also positively correlated with Skindez 29-symptoms, emotional and functioning [38]. Nevertheless, it was observed that NRS for suppuration did not have an impact on sex life (β = 0.05, *p* = 0.489) [25], neither in men nor in women [25].

## 4. Discussion

The results of this systematic review and meta-analysis presents the clinical situation of patients with HS regarding signs and symptoms and summarizes the current evidence regarding their correlation with QoL impairment, both general and specific. The importance of the research on this topic is notable and increasing, as the majority of the studies are published from 2016 onwards.

As previously described, HS has a great impact on QoL [41], even more than other dermatosis, such as psoriasis or atopic dermatitis [42]. Although most tools used to assess QoL were validated questionnaires, they differed between studies. The tool most frequently used to evaluate QoL was the DLQ, showing moderate to large impacts on patients’ lives [26,27,28,30,32,33,35,36]. The scores employed to assess signs and symptom severity were subjective and varied between studies, with the NRS being the most used tool for symptoms [24,32,33]. Furthermore, it is noteworthy that clinicians from North America and Asia are less likely to measure HS symptoms, which may reflect regional differences in clinical assessment or research trends [43].

Pain is the symptom with the strongest correlation with QoL impairment [7,44,45]. Mean pain reported was almost four out of 10, which qualifies as mild-to-moderate pain considering established cut-offs [46]. These values are like chronic posttraumatic headaches and worse than vasculitis, blistering disorders, vulvar lichen sclerosis and leg ulcers [19,47]. HS pain is both nociceptive and neuropathic. Nociceptive pain may be the result of acute inflammation while neuropathic HS pain could be due to chronic inflammation causing peripheral neuroplastic changes and central sensitization. Addressing HS pain is critical to improve HS-related QoL and reduce morbidity from opioid and other substance use. Unfortunately, current HS therapies often provide inadequate pain relief, and studies of HS pain-directed therapies are sparse. Non-steroidal anti-inflammatory drugs, intralesional corticosteroids or neuromodulator medications could be effective treatment for pain [44]. Moreover, incorporation of psychological therapies may represent an important opportunity for reducing chronic HS pain [43]. Pain intensity correlated with impairment in QoL in all the studies included [24,25,26,27,28,29,30,31,32,33,34,35,36]. This is in part explained by the physical limitations caused by the painful lesions. Moreover, pain is associated with poor mental health. Rates of depression and anxiety are higher in HS patients than in healthy individuals [48,49,50]. Pro-inflammatory cytokines, including TNF- α, IL-1β and IL-10, are elevated in the lesional skin of HS patients [49,51]. TNF-α and IL-1β are also increased in major depressive, anxiety and other psychiatric disorders [52]. Therefore, high levels of these cytokines in HS [49,53] could explain the relationship between HS and poor mental health. HS also has an impact on sleep quality, even worse than other systemic conditions, such as lupus erythematosus, chronic obstructive pulmonary disease or Hodgkin’s lymphoma [54,55]. Sleep disorders also contribute to decreased QoL [56] and HS pain impact on sleep quality [29]. Sexual health is likewise an important aspect of patients’ QoL [57] and pain is also a risk factor for sexual distress and sexual dysfunction [24,25]. Sexual distress reveals the suffering of the subject while sexual dysfunction might mean a poor sexual experience for both members of the relationship [25]. The impact of pain in sexual health may be linked to the nature of the sexual act and psychological factors that may be associated with disease activity [57].

Pruritus is the second symptom with the strongest correlation with poorer QoL [24,25,29,32,33,34,39,40]. It might be underreported because patients do not spontaneously refer to this symptom unless they are specifically asked [32]. The mean pruritus reported was almost five out 10. Although HS is not considered as a pruritic disease, this symptom is commonly associated with HS, mainly during the outbreak of lesions [20,58]. Pruritus severity has been related to overall impairment of QoL [32,33,34,39], sleep disturbances [29,40] and depression [34] but it has not been linked to poor sexual health [24,25]. In agreement with our results, pruritus has been previously described as a potential risk factor for sleep impairment in other dermatosis [59,60]. The absence of impact on sexual health might be due to pruritus being less bothersome than pain or it could even decrease during sexual intercourse. Skin irritation caused by suppuration might be the cause of pruritus in patients with HS [33]. The increased number of mast cells and inflammatory cell infiltration in HS lesions might also explain the pruritus in HS [7,32]. The reduction of suppuration through antibiotics, anti-inflammatories, or surgical procedures, as well as topical measures to control skin irritation like emollients or corticosteroids, should be considered in patients with the relevant pruritus and suppuration [33].

Scarce studies have evaluated the impact of malodour and suppuration on QoL [24,25,33,37,38,39]. Compared to pain and pruritus, malodour and suppuration can potentially be perceived by other people and might also contribute to worsen patients QoL and stigmatization. In fact, malodour can be underestimated by the patients as they get used to it, but their partners usually show a more expressive response when they inquire about this problem. This can potentially cause social, work and personal problems, and favours stigma and isolation behaviours [33,61], contributing to decreased QoL. Studies have also shown that malodour and suppuration severity were associated with poorer sexual QoL only in women, not in men [24]. Although previous investigations indicated higher sexual distress in women than in men with HS, the impact of different symptoms between both sexes might be explained by an early onset of HS in women, or even by cultural aspects and differences in emotional and neuroendocrine response to disfigurement [62]. As previously stated, suppuration is linked to advanced chronic lesions and bacterial biofilms [63,64]. Intensity of suppuration, Hurley stage, longer disease duration and high number of regions affected are risk factors for malodour [33]. Structural damage, as in the presence of scars, may make personal hygiene difficult and favour bacterial overgrowth thereby increasing molodour and suppuration scores. Body mass index is associated with malodour probably due to the presence of prominent skin folds and excessive sweating. Control of malodour and suppuration should be a priority in patients with HS to improve their QoL. Weight loss is advisable in all overweight and obese patients. Anti-inflammatories, antibiotics or a combination of both should be given in patients with poor disease control, and antiseptic washes or surgical procedures to remove scarring tissue could be used in patients with good disease control and structural damage [33].

This systematic review is subject to some limitations. All the designs were cross-sectional, which limits the inference of causality. Although most of the questionnaires and tools used are validated, there is heterogeneity between the different studies. There are also differences in the severity of the patients depending on the source of the patients (outpatient clinic vs. general surveys). Moreover, some the assessment of these symptoms is subjective, which may also increase the variability.

Further studies should include validating questionnaires to assess QoL and symptoms severity. DLQI might be a good option to evaluate overall QoL. NRS for pain, pruritus, malodour and suppuration should also be included. Outcomes should be reported, both cut-off (nominal) and average (continuous) data of these questionnaires, in future studies.

## 5. Conclusions

In conclusion, we have summarized, categorized and analyzed the scientific evidence regarding signs and symptoms in HS patients and their impairment in QoL. Pain might be the symptom most related with impairment in QoL due to its high frequency and subjective component. Malodour is the least studied symptom and could have a major effect on interpersonal relationships. Assessment of these symptoms should be thorough and included during routine evaluation of HS patients. It would be important to define cut-off values of symptom severity to motivate therapeutic modifications. Coordinated and consistent medical and psychological support are of great importance to increase patients’ health.

## Figures and Tables

**Figure 1 ijerph-18-06709-f001:**
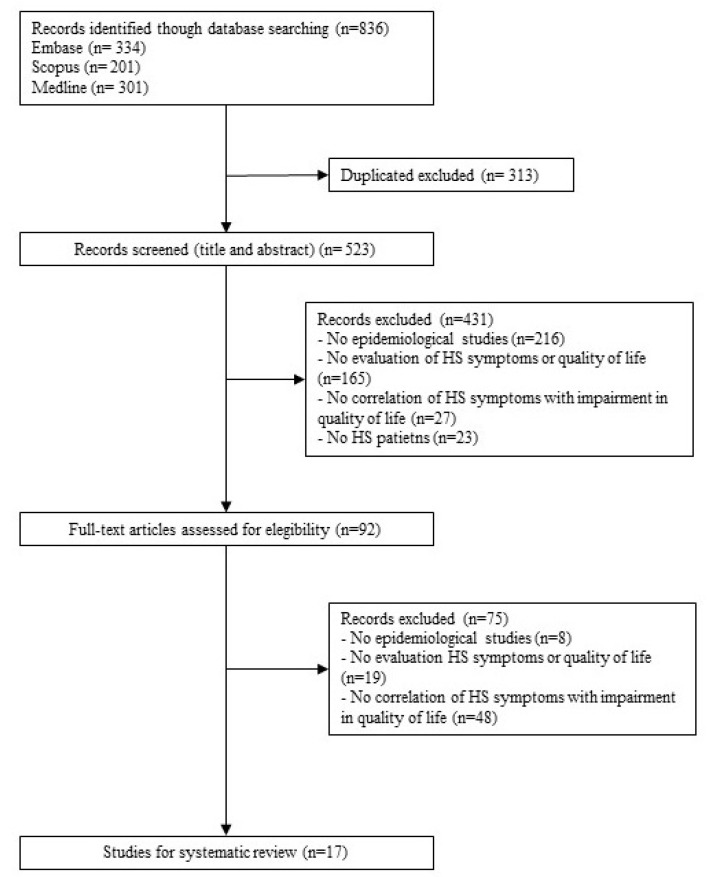
Flow chart of the studies included.

**Figure 2 ijerph-18-06709-f002:**
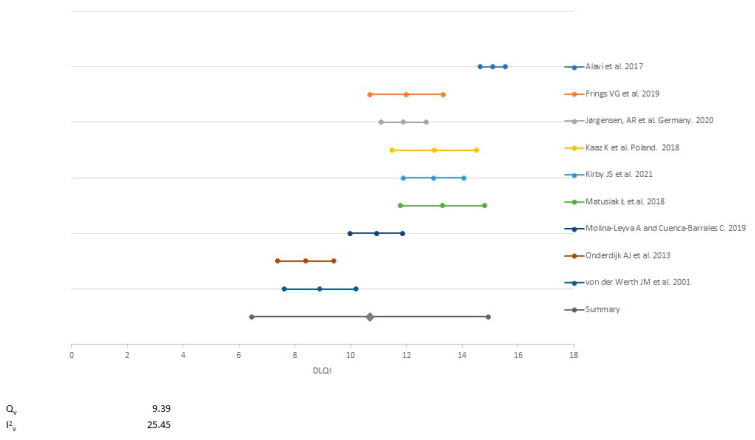
Meta-analysis of the mean Dermatology Life Quality Index reported in the studies.

**Figure 3 ijerph-18-06709-f003:**
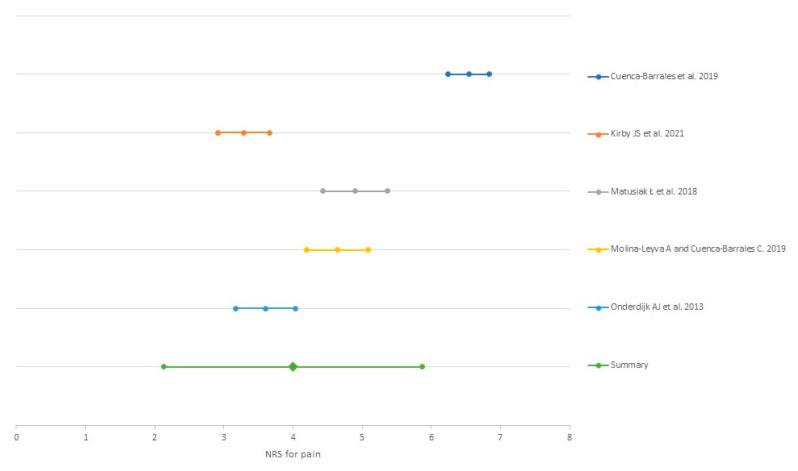
Meta-analysis of the mean Numerical Rating Scale for pain reported in the studies.

**Figure 4 ijerph-18-06709-f004:**
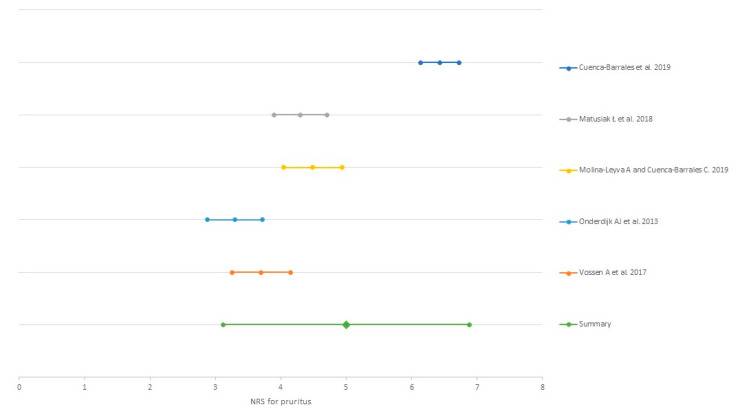
Meta-analysis of the mean Numerical Rating Scale for pruritus reported in the studies.

**Table 1 ijerph-18-06709-t001:** Main characteristics of the studies regarding HS symptoms impact on QoL.

Study, Site and Year	Design	CEBM	Participants	Age (Years)	Sex (Female:Male Ratio)	BMI (kg/m^2^)	Smoking Habit (Yes)	Disease Duration (Years)	Disease Severity (Hurley I/II/III)	HS Symptom Evaluated	Main Aspects Od QoL Evaluated
Alavi et al. Canada 2017 [37]	Cross-sectional	4	51	32.50 ± 10.76	2.47:1	NS	NS	11.10 ± 8.57	13.7% (7)/45.1% (23)/41.2% (21)	Malodour	Overall QoL
Cuenca-Barrales et al. Spain. 2019 [25]	Cross-sectional	4	386	37.81 ± 9.26	3.83:1	29.35 ± 6.71	42.2% (163)	17.77 ± 9.62	17.6% (68)/45.1% (174)/37.3% (144)	Pain, pruritus, malodour, suppuration	Sexual distress
Cuenca-Barrales and Molina-Leyva. Spain. 2020 [24]	Cross-sectional	4	386	37.81 ± 9.26	3.83:1	29.35 ± 6.71	42.2% (163)	17.77 ± 9.62	17.6% (68)/45.1% (174)/37.3% (144)	Pain, pruritus, malodour, suppuration	Sexual dysfunction
Frings et al. Germany. 2019 [26]	Cross.sectional	4	110	38 ± 12	1.24:1	30.5 ± 6.9	37% (41)	NS	7% (8)/30% (33)/63% (69)	Pain	Overall QoL, anxiety and depression
Huilaja et al. Finland. 2020 [27]	Cross-sectional	4	92	NS *	NS *	NS *	NS *	NS *	NS *	Pain	Overall QoL and depression
Jørgensenet al. Germany. 2020 [28]	Cross-sectional	4	339	39.4 ± 13.5	1.80:1	29 ± 7.6	75.2% (255)	13.8 ± 11.5	28.3% (96)/ 57.5% (195)/14.2% (48)	Pain	Overall QoL
Kaaz et al. Poland. 2018 [29]	Cross-sectional	4	108	36.3 ± 12.1	0.89:1	28.8 ± 5.4	18% (60)	9.1 ± 8.3	46.3% (50)/45.4% (49)/ 8.3% (9)	Pain and pruritus	Sleep and insomnia
Kirby et al. Denmark and USA. 2021 [30]	Cross-sectional	4	224	39.6 (19-77)	6.72:1	NS	NS	NS	NS	Pain	Overall QoL
Krajewski et al. Germany and Poland. 2021 [31]	Cross-sectional	4	1795	40.0 ± 11.8	1.79:1	28.1 ± 6.2	55.6% (998)	NS	NS	Pain	Overall QoL
Machado et al. Canada. 2021 [38]	Cross-sectional	4	30	40.87 ± 2.55	2.33:1	NS	NS	NS	NS	Malodour and drainage	Overall QoL
Matusiak et al. Poland 2018. [32]	Cross-sectional	4	103	35.6 ± 13.2	0.94:1	29.4 ± 5.7	54.4% (56)	8.9 ± 7.5	40.8% (42)/45.6% (47)/13.6% (14)	Pain and pruritus	Overall QoL
Molina-Leyva and Cuenca-Barrales. Spain. 2019 [33]	Cross-sectional	4	233	40.14 ± 13.46	1.14:1	30.68 ± 7.05	NS	13.99 ± 10.59	30.04% (70); 41.63% (97); 28.33% (66)	Pruritus, malodour	Overall QoL
Onderdijk et al. Netherlands. 2013 [34]	Cross-sectional	4	211	43.0 ± 11.8	NS	NS	NS	16.8 ± 11.6	30.1% (64)/56.4% (119)/13.5% (28)	Pain and pruritus	Overall QoL and depression
Riis et al. Denmark. 2016 [39]	Cross-sectional	4	421	42.4 (19-77)	3.74:1	NS	NS	NS	NS	Pain, pruritus and malodour	Health-realted overall utility
Sampogna et al. Italy 2019 [35]	Cross-sectional	4	69	34.5 ± 12.5	1.16:1	NS	NS	NS	27.5% (19)/43.5% (30)/29.0% (20)	Pain	Overaal QoL
von der Werth et al. Denmark. 2001 [36]	Cross-sectional	4	160	40.9 ± 11.7	6.13:1	NS	NS	NS	NS	Pain	Overall QoL
Vossen et al. Netherlands. 2017 [40]	Cross-sectional	4	211	38.0 (29–49)	1.78:1	28.5 ± 5.9	19.9% (62)	14.0 (7–25)	15% (32)/66% (140)/19% (39)	Pruritus	Activities of daily living and sleep

BMI, body mass index; CEBM, level of scientific evidence according to the Centre for Evidence-Based Medicine (24); DLQI, Dermatology Life Quality Index; HADS, Hospital Anxiety and Depression Scale; HS, hidradenitis suppurativa; HSS, Hidradenitis Suppurativa Score; IHS4, International Hidradenitis Suppurativa Severity Index; NRS, Numeric Rating Scale; NS, not specified; QoL, quality of life, VAS, visual analogue scale. Continuous data is expressed as media ± SD or median (interquartile range) and categorical data are presented as n or n/N (%). * The article contains this information in the Appendix A but it is not accessible (broken link).

**Table 2 ijerph-18-06709-t002:** Studies regarding pain impact on QoL.

Study	Pain		QoL			Correlation between Pain and QoL
	Assessment Tool	Score	Assessment Tool		Score	
Cuenca-Barrales et al. Spain. 2019 [25]	NRS	6.54 ± 2.95	NRS for HS impact on sex life		7.24 ± 2.77 in women 6.39 ± 3.44 in men	β = 0.15, *p* = 0.049 **
Cuenca-Barrales and Molina-Leyva. Spain. 2020 [24]	NRS	Women: 6.52 ± 2.98	Prevalence of sexual dysfunction	FSFI-6 ≤ 19	51% (156)	β = 0.1, *p* < 0.05 **
		Men: 6.64 ± 2.81		IIEF-5 ≤ 21	60% (48)	β = NS, *p* = 0.97 **
Frings et al. Germany. 2019 [26]	VAS	NS	DLQI		12 ± 7	r = 0.457, *p* < 0.001
			HADS-Depression		6 ± 4	r = 0.193, *p* = 0.105
			HADS-Anxiety		7 ± 4	r = 0.304, *p* = 0.009
			Skindex-29 symptom score		NS	r = 0.547, *p* < 0.001
			Skindex-29 function score			r = 0.459, *p* < 0.001
			Skindex-29 emotion score			r = 0.399, *p* < 0.001
Huilaja et al. Finland. 2020 [27]	VAS: -No pain (0–4 mm)/ -Mild pain (5–44 mm)/ -Moderate to severe pain (45–100 mm)	37% (34)/ 45.7% (42)/ 17.4% (16)/	DLQI		3.03 (0–9) vs. 8.76 (0–23) vs. 13.69 (4–29)	*p* < 0.001 ^^¶^^
			Beck’s Depression Inventory		6.68 (0–4.0) vs. 9.26 (0–30) vs. 13.06 (1–32)	*p* = 0.019 ^^¶^^
	PainDETECT: -Pain negative (0–12)/ -Unclear (13–18)/ -Pain positive (19–38)	41.3% (38)/ 27.2% (25)/ 31.5% (29)	DLQI		4.53 (0–16) vs. 8.84 (1–23) vs. 10.55 (0–29)	*p* < 0.001 ^^¶^^
			Beck’s Depression Inventory		6.84 (0–20) vs. 7.68 (0–19) vs. 12.86 (0–32)	*p* = 0.003 ^^¶^^
Jørgensen et al. Germany. 2020 [28]	Boil-associated pain score (0–10) ≤5 boils/ >5 boils	8.6 ± 7.4/ 15 ± 7.4	DLQI		11.9 ± 7.6	Mean difference: 6.3 ± 1.85, *p* < 0.001 ʡ
Kaaz et al. Poland. 2018 [29]	VAS	4.9 ± 2.9	DLQI		13.0 ± 8.0	NS
			AIS		5.4 ± 4.3	*p* < 0.05 ^
			PSQI		6.5 ± 3.6	*p* < 0.05 ^
Kirby et al. Denmark and USA. 2021 [30]	NRS	3.29 ± 2.83	DLQI		12.97 ± 8.33	NS
			PtGA of QoL (0–4)		2.09 ± 1.34	r = 0.66 (0.6–0.71 95% CI) #
Krajewski et al. Germany and Poland. 2021 [31]	NRS	NS	DLQI		13.2 ± 8.1	r = 0.581; *p* < 0.001 ^
Matusiak et al. Poland. 2018 [32]	Prevalence of pain	77.5% (80)	DLQI		13.3 ± 7.8	NS
	VAS	4.6 ± 2.5				r = 0.48, *p* < 0.001 ^^
	NRS	4.9 ± 2.4				r = 0.48, *p* < 0.001 ^^
Molina-Leyva and Cuenca-Barrales. Spain. 2019 [33]	NRS	4.64 ± 3.43	DLQI		10.93 ± 7.3	β = 0.91 ± 0.12, R2 = 0.36, *p* < 0.001 *
Onderdijk et al. Netherlands. 2013 [34]	NRS	3.6 ±3.2	DLQI		8.4 ± 7.5	r = 0.60, *p* < 0.05 ʡ
			MDI		4.3 ± 5.6	r = 0.36, *p* < 0.05 ʡ
Sampogna et al. Italy. 2019 [35]	VAS: <5/ 5–6/ ≥7	25.9% (11)/ 29% (20)/ 55.1% (38)	Skindex-17 Symptoms		53.6 vs. 72.0 vs. 72.6,	*p* = 0.068^¶^
			Skindex-17 psychosocial		39.4 vs. 54.6 vs. 61.7	*p* = 0.088 ^^¶^^
von der Werthet al. Denmark. 2001 [36]	Self-reported number of painful lesions	2	DLQI		8.9 ± 8.3	r = 0.384, *p* < 0.01 ʡ

AIS, Athens Insomnia Scale; DLQI, Dermatology Life Quality Index; FSFI-6, six-item Female Sexual Function Index; HADS, Hospital Anxiety and Depression Scale; HS, hidradenitis suppurativa; IIEF-5, five-item International Index of Erectile Function; MDI, Major Depression Inventory; NRS, Numeric Rating Scale; NS, not specified; PSQI, Pittsburgh Sleep Quality Index; PtGA, patient global assessment; QoL, quality of life. * Simple linear regression analysis; ** Multivariate linear regression analysis; Analysis of variance; # Student’s *t*-test for independent samples; ^¶^ Multivariate analysis of variance; ʡ Spearman correlation; ^ Pearson correlation; ^^ Pearson’s correlation coefficient or Spearman’s correlation analysis with reference to the distribution of evaluated variables.

**Table 3 ijerph-18-06709-t003:** Studies regarding pruritus impact on QoL.

Study	Pruritus		QoL			Correlation between Pruritus and QoL	Factors Associated with Pruritus
	Assessment Tool	Score	Assessment Tool		Score		
Cuenca-Barrales et al. Spain. 2019 [25]	NRS	6.43 ± 2.96	NRS for HS impact on sex life		7.24 ± 2.77 in women and 6.39 ± 3.44 in men	β = 0.03, *p* = 0.615 **	NS
Cuenca-Barrales and Molina-Leyva. Spain. 2020 [24]	NRS	Women: 6.48 ± 3.03	Prevalence of sexual dysfunction	FSFI-6 ≤ 19	51% (156)	β = NS, *p* = 0.36 **	NS
		Men: 6.24 ± 2.67		IIEF-5 ≤ 21	60% (48)	β = NS, *p* = 0.28 **	NS
Kaaz et al. Poland. 2018 [29]	VAS	4.1 ± 2.9	DLQI		13.0 ± 8.0	NS	NS
			AIS		5.4 ± 4.3	*p* < 0.05 ^	
			PSQI		6.5 ± 3.6	*p* < 0.05 ^	
Matusiak et al. Poland. 2018 [32]	Prevalence of pruritus	41.7%	DLQI		13.3 ± 7.8	*p* = 0.79^¶^	Hurley III, active smokers ^^¶^^
	VAS	3.9 ± 2.2				r = 0.45, *p* = 0.004 ^^	
	NRS	4.3 ±2.1				r = 0.48, *p* = 0.002 ^^	
	4-item Itch Questionnaire	4.6 ± 1.9				NS	
Molina-Leyva and Cuenca-Barrales. Spain. 2019 [33]	NRS (NRS pruritus> 3)	4.49 ± 3.48 (61.8% (144))	DLQI (DLQI> 10)		10.93 ± 7.3 (49.79% (119))	β = 0.42 ± 0.11, *R*2 = 0.20, *p* < 0.001 *	Number of regions affected (β = 0.51, *p* = 0.01), female sex (β = 0.46, *p* = 0.02), intensity of suppuration (β = 0.42, *p* < 0.001), Crohn’s disease (β = 1.24, *p* = 0.01), not statin use (β= 0.87, *p* = 0.03) #
Onderdijk et al. Netherlands. 2013 [34]	NRS	3.3 ± 3.1	DLQI		8.4 ± 7.5	r = 0.53, *p* < 0.05 ʡ	NS
			MDI		4.3 ± 5.6	r = 0.33, *p* < 0.05 ʡ	
Riis et al. Denmark. 2016 [39]	NRS	NS	EQ-5D		NS	β= −0.017, *p* < 0.05 **	NS
Vossen et al. Netherlands. 2017 [40]	Prevalence of pruritus (NRS score ≥3)	57.3% (121)	ADL Sleep		70% (36/51) 53% (27/51)	NS	Hurley stage III (OR 7.73; *p* = 0.003) and higher levels of pain, (OR = 1.34 for each additional point on the NRS, *p* = 0.001) #
	NRS	3.7 ± 3.3					
	5-D itch scale	13.7 ± 3.6					

ADL, activities of daily living; DLQI, Dermatology Life Quality Index; EQ-5D, EuroQoL-5D; FSFI-6, six-item Female Sexual Function Index; HS, hidradenitis suppurativa; IIEF-5, Five-Item International Index of Erectile Function; MDI, Major Depression Inventory; NRS, Numeric Rating Scale; NS, not specified; QoL, quality of life. * Simple linear regression analysis; **## Multivariate linear regression analysis; # Multivariate logistic regression analysis; ^¶^ Multivariate analysis of variance; ʡ Spearman correlation; ^ Pearson correlation; ^^ Pearson’s correlation coefficient or Spearman’s correlation analysis with reference to the distribution of evaluated variables.

**Table 4 ijerph-18-06709-t004:** Studies regarding malodour impact on QoL.

Study	Malodour		QoL Score			Correlation between Malodour and QoL	Risk Factors for Malodour
	Assessment Tool	Score	Assessment Tool		Score		
Alavi et al. Canada 2017 [37]	NRS	5.02 ± 3.06	DLQI		15.10 ± 1.64	*R*2 = 0.17, F = 2.63, *p* = 0.064 ^a^ **	Lesions groin, upper thighs, and buttocks (Pearson χ^2^ = 5.66, df = 1, *p* = 0.017).
			Skindex-19		65.33 ± 17.18	*R*2 = 0.39, F = 8.11, *p* < 0.001 ^a^ **	
Cuenca-Barrales et al. Spain. 2019 [25]	NRS	5.6 ± 3.38	NRS for HS impact on sex life		7.24 ± 2.77 in women and 6.39 ± 3.44 in men	β = 0.13, *p* = 0.035 **	NS
Cuenca-Barrales and Molina-Leyva. Spain. 2020 [24]	NRS	Women: 5.47 ± 3.45	Prevalence of sexual dysfunction	FSFI-6 ≤ 19	51% (156)	β = 0.07, *p* < 0.05 **	NS
		Men: 6.11 ± 3.05		IIEF-5 ≤ 21	60% (48)	β = NS, *p* = 0.52 **	NS
Machado et al. Canada. 2021 [38]	HODS-odour:	NS	Skindex-29		NS	r = 0.726, *p* < 0.05	NS
			Skindex-29 symptoms		NS	r = 0.733, *p* < 0.05 ^	
			Skindex-29 emotional		NS	r = 0.725, *p* < 0.05 ^	
			Skindex-29 functioning		NS	r = 0.719, *p* < 0.05 ^	
			HS-QoL overall		NS	r = 0.719, *p* < 0.05 ^	
Molina-Leyva and Cuenca-Barrales. Spain. 2019 [33]	NRS (NRS > 3)	3.28 ± 3.58 (40.77% (95))	DLQI (DLQI > 10)		10.93 ± 7.3 (49.79% (119))	β = 0.44 ± 0.11, *R*2 = 0.23, *p* < 0.001 *	Higher BMI (β = 0.04, *p* = 0.07), Disease duration (β = 0.05, *p* = 0.01), Number of regions affected (β = 0.31, *p* = 0.08), Hurley stage (β = 0.60, *p* = 0.02), intensity of suppuration (β = 0.61, *p* < 0.001)
Riis et al. Denmark. 2016 [39]	NRS	NS	EQ-5D		NS	β= −0.023, *p* < 0.05 **	NS

BMI, body mass index; DLQI, Dermatology Life Quality Index; EQ-5D, EuroQoL-5D; FSFI-6, six-item Female Sexual Function Index; HS, hidradenitis suppurativa; HODS, Hidradenitis Suppurativa Odor and Drainage Scale; HS-QoL, Hidradenitis Suppurativa quality-of-life measure; IIEF-5, five-item International Index of Erectile Function; NRS, Numeric Rating Scale; NS, not specified; QoL, quality of life. ^a^ Correlation are done between malodour severity and QoL assessment tools. * Simple linear regression analysis; ** Multivariate linear regression analysis; # Multivariate logistic regression analysis; ^ Pearson correlation.

**Table 5 ijerph-18-06709-t005:** Studies regarding suppuration impact on QoL.

Study	Suppuration		QoL Score			Correlation between Suppuration and QoL
	Assessment Tool	Score	Assessment Tool		Score	
Cuenca-Barrales et al. Spain. 2019 [25]	NRS	6.48 ± 3.18	NRS for HS impact on sex life		7.24 ± 2.77 in women and 6.39 ± 3.44 in men	β = 0.05, *p* = 0.489 **
Cuenca-Barrales and Molina-Leyva. Spain. 2020 [24]	NRS	Women: 6.39 ± 3.21	Prevalence of sexual dysfunction	FSFI-6 ≤ 19	51% (156)	β = NS, *p* = 0.29 **
		Men: 6.48 ± 3.04		IIEF-5 ≤ 21	60% (48)	β =NS, *p* = 0.98 **
Machado et al. Canada. 2021 [38]	HODS-drainage:	NS	Skindex-29		NS	r = 0.614, *p* < 0.05 ^
			Skindex-29 symptoms		NS	r = 0.619, *p* < 0.05 ^
			Skindex-29 emotional		NS	r = 0.616, *p* < 0.05 ^
			Skindex-29 functioning		NS	r = 0.605, *p* < 0.05 ^
			HS-QoL overall		NS	r = 0.745, *p* < 0.05 ^

FSFI-6, six-item Female Sexual Function Index; HODS, Hidradenitis Suppurativa Odor and Drainage Scale; HS, hidradenitis suppurativa; HS-QoL, Hidradenitis Suppurativa quality-of-life measure; IIEF-5, five-item International Index of Erectile Function. ^ Pearson correlation; ** Multivariate linear regression analysis.

## Data Availability

The data presented in this study are available on request from the corresponding author.

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
