# Peer review of "The Burden of Hidradenitis Suppurativa Signs and Symptoms in Quality of Life: Systematic Review and Meta-Analysis"

_ijerph, 2021, doi:10.3390/ijerph18136709_

Round 1

Reviewer 1 Report

This is an interesting study worth publishing. I like the methodology employed and the style of presenting the results. My comments are as follows:

  1. As this is a mata-analysis it should be registered in PROSPERO database. Please comment on this issue.
  2. Did the authors assess the risk of bias of 17 selected articles?
  3. Dr Krajewski is from Poland, not Germany. This should be corrected in table 1.
  4. Why the country is added only in table 1 and in some citations in table 2? The style should be the same for all the tables.
  5. I miss the paper by Krajwski et. on pain in HS publsihed recently in Acta Dermatovenereologica.
  6. Introduction: I propose to add the specific tools to asssess QoL which are now availbale and have been used in some studies. One intrument was even created in Spain by the group from Zaragosa.

Author Response

Reviewer 1

This is an interesting study worth publishing. I like the methodology employed and the style of presenting the results. My comments are as follows:

Thank you for the comments.

As this is a mata-analysis it should be registered in PROSPERO database. Please comment on this issue.

Thank you for your advice. As it is advisable but not compulsory, we had not registered this meta-analysis previously in PROSPERO. Following your wise recommendation, it has been now registered with ID 261476 and we will perform this task in future meta-analyses.

Did the authors assess the risk of bias of 17 selected articles?

Thank you for your comment. All the articles available and included has a cross-sectional design, with similar quality and risk of bias. Following your suggestion, we have included a new section in material and methods an in the results section with a new figure assessing the risk of biases using The National Institutes of Health quality assessment tool.

Dr Krajewski is from Poland, not Germany. This should be corrected in table 1.

Nationality was indicated based in the country where the patients were recruited. We believe that this has more important implications regarding symptoms and disease experience. If you consider that is better to mention the first author country, we can change it.

Why the country is added only in table 1 and in some citations in table 2? The style should be the same for all the tables.

Sorry for the mistake. We have included the country in all the tables.

I miss the paper by Krajwski et. on pain in HS publsihed recently in Acta Dermatovenereologica.

We did not include it because the results were the same that those Krajewski et al published in Life (mean DLQI: 13.2+-8.1 points; QoL impairment correlated positively with pain (r = 0.581, p < 0.001). If we had included both of them the population will be artificially duplicated and will bias the meta-analysis. After analyzing both articles, the one published in Life focus more specifically on QoL and we decided to include it instead of the one ActaDV. Following your recommendation, the article of ActaDV has been now included in the discussion section.

Introduction: I propose to add the specific tools to asssess QoL which are now availbale and have been used in some studies. One intrument was even created in Spain by the group from Zaragosa

We have included the specific tools available for assessing QoL in HS patients. The following sentence has been added in the introduction: Recently, disease specific instruments to assess quality of life in HS has been developed such as the HSQoL-24 validated in the Spanish population.

Reviewer 2 Report

The manuscript entitled „The burden of hidradenitis suppurativa signs and symptoms in quality of life: systematic review and meta-analysis” presents interesting issues but the manuscript should be corrected.

Editing:

The manuscript is shabbily prepared and should be corrected (e.g. missing spaces or redundant spaces, tables not prepared according to instructions for authors, etc.).

Figures should be presented within the manuscript, not as a supplementary material, as they are the most important element for the conducted meta-analysis.

Abstract:

Authors should not begin with the statement indicating the influence of HS on the quality of life, as it is to be studied within the manuscript and indicating it as a background may suggest that the study was not needed as we know everything about it.

Authors should present here any basic characteristics of patients presented in included studies, e.g. mean age in the studies differing from… to…, general proportion of male and female participants in the studies, general assessment of BMI, etc.

Introduction:

Authors should not formulate definitive statements associated with the objective of the study, as it is to be studied within the manuscript and indicating it as a background may suggest that the study was not needed as we know everything about it (e.g. „HS is one of the dermatological disease with the greatest impact on patients’ quality of life (QoL)”;  „Its impairment is similar to other conditions such as cardiovascular disease, cancer, diabetes mellitus and chronic obstructive pulmonary disease”). Authors especially can not formulate such statements based on only 1 study referred, as their systematic review and meta-analysis should be published to provide valid information about it, so they can not suggest that we know everything about it. Instead, AUthors shpuld rather present any inconsistent results in the single studies (if they exist), to indicate, that a systematic review is needed.

Materials and Methods:

Authors should indicate on the basis of what assumption they created the list of symptoms to be included („pain or itch or odour or malodour or suppuration or oozing or drainage”) – e.g. specific reference defining the symptoms.

What do Authors mean by „in vivo”? – if Authors intended to include only human studies, „human studies” is enough, if not - they should specify what do they mean to include while indicating „in vivo” – e.g. animal model studies (if so, what for?)

Authors should precisely describe what data were extracted from the studies

Authors should precisely decribe applied methodology of meta-analysis

Authors should assess their included studies in any way, e.g. while using Newcastle-Ottawa Scale (NOS)

Results:

In the tables each referred study should be presented with a proper reference (the same as in the text

Authors should assess their included studies in any way, e.g. while using Newcastle-Ottawa Scale (NOS)

Discussion:

Authors should also indicate other limitations of the study assciated e.g. with the numer of the studies conducted so far which were possible to be included, or subjective character of the data.

Conclusions:

Authors should try to formulate any specific conclusion, associated e.g. with the symptom influecing quality of life in the most signifficant way.

Authors Contributions:

Authors should present specific actions which were conducted within the study – e.g. what do Authors mean by „validation”, if there is no validation presented?

It seems that majority of Authors did not present in the manuscript preparation. It is a serious risk of the guest authorship procedure, which is forbidden. They should be either not included as authors (and presented only in Acknowledgements section), or their participation in manuscript preparation should be indicated.

Author Response

Reviewer 2

The manuscript entitled „The burden of hidradenitis suppurativa signs and symptoms in quality of life: systematic review and meta-analysis” presents interesting issues but the manuscript should be corrected.

Thank you for the comments

 Editing:

The manuscript is shabbily prepared and should be corrected (e.g. missing spaces or redundant spaces, tables not prepared according to instructions for authors, etc.).

We have thoroughly reviewed the manuscript to avoid editing problems. We have also changed the format of the tables as recommended.

Figures should be presented within the manuscript, not as a supplementary material, as they are the most important element for the conducted meta-analysis.

Figures have been presented within the manuscript as recommended instead of as supplemented material

Abstract:

Authors should not begin with the statement indicating the influence of HS on the quality of life, as it is to be studied within the manuscript and indicating it as a background may suggest that the study was not needed as we know everything about it.

We have changed the introduction of the abstract and have not focus on QoL impairment. We have described HS disease and explained its signs and symptoms. The following text has been added Hidradenitis suppurativa (HS) is a chronic, recurrent and debilitating inflammatory skin disease of the hair follicle that usually presents painful, deep-seated inflamed lesions in the apocrine gland-bearing areas of the body. HS patients suffer from uncomfortable signs and symptoms, such as pain, pruritus, malodor and suppuration, which may impair patients’ quality of life (QoL).

Authors should present here any basic characteristics of patients presented in included studies, e.g. mean age in the studies differing from… to…, general proportion of male and female participants in the studies, general assessment of BMI, etc.

 We have included information regarding basic characteristic of patients in the studies (age, sex, BMI, smoking habit, HS severity). The following sentences have been added: Mean age of the participants was 36.28 years and there was a predominance of female sex among study participants. The BMI of the population was in the range of over-weight and about 2 out 5 patients was active smoker. Studies included patients with mild to moderate HS, with a mean disease duration of 13.69 years.

Introduction:

Authors should not formulate definitive statements associated with the objective of the study, as it is to be studied within the manuscript and indicating it as a background may suggest that the study was not needed as we know everything about it (e.g. „HS is one of the dermatological disease with the greatest impact on patients’ quality of life (QoL)”;  „Its impairment is similar to other conditions such as cardiovascular disease, cancer, diabetes mellitus and chronic obstructive pulmonary disease”). Authors especially can not formulate such statements based on only 1 study referred, as their systematic review and meta-analysis should be published to provide valid information about it, so they can not suggest that we know everything about it. Instead, AUthors shpuld rather present any inconsistent results in the single studies (if they exist), to indicate, that a systematic review is needed.

Thank you for your comment. It is known that HS impair QoL as it is mentioned in the text. Nevertheless, it is unknown how this HS symptoms impact on QoL. There are inconsistences regarding which is the most frequent and most annoying symptom. In our study we are not assessing a rate of DLQI, NRS for pain, NRS for pruritus, etc. We are evaluating in what extent Pain Intensity (NRS PAIN) impacts on quality of life (DLQI) for example. This is the spot, where scientific evidence is needed. Following your recommendations, we have modified the following sentence Pain, pruritus, malodour and suppuration are signs and symptoms frequently experienced by patients, but they are only occasionally assessed by clinicians. Signs and symptoms might be the main burden of patients with HS, producing a great impairment in quality of life. Unexpectedly, the scientific evidence available is limited and heterogeneous. The aim of this study is to summarize the evidence regarding the impact of HS signs and symptoms on QoL to serve as a basis for future research and help clinicians to consider them in the daily care of HS patients.

Materials and Methods:

Authors should indicate on the basis of what assumption they created the list of symptoms to be included („pain or itch or odour or malodour or suppuration or oozing or drainage”) – e.g. specific reference defining the symptoms.

Symptoms included in the literature search were selected by a dermatologist expert in HS (AML) following the most recent evidence in HS clinical presentation. This sentence has been added in the material and methods section.

What do Authors mean by „in vivo”? – if Authors intended to include only human studies, „human studies” is enough, if not - they should specify what do they mean to include while indicating „in vivo” – e.g. animal model studies (if so, what for?)

We mean that we only included human studies. The sentence regarding in vivo studies has been omitted.

Authors should precisely describe what data were extracted from the studies

The data extracted from the studies were number of participants, age, sex, HS symptoms and aspects of QoL evaluated, QoL and symptoms assessment tools and scores. This is included in the material and methods sections.

Authors should precisely decribe applied methodology of meta-analysis

We use the material provided by Neyeloff et al and followed their instruction to conduct the meta-analysis. The mean DLQI and NRS for symptoms was calculated by a random effect meta-analysis weighted by the study sample size. Forest plots were constructed to summarize the effect size and their 95% CIs. These figures present measures of heterogeneity across studies (Cochrane Q statistic, noted the I2 statistic). Microsoft Excel version 2016, Redmond, Washington, The USA. was used to run this data. This information is included in the material and methods section.

Authors should assess their included studies in any way, e.g. while using Newcastle-Ottawa Scale (NOS)

 As all the studies were cross-sectional we used the National Institutes of Health quality assessment tool to evaluate risk of bias. This information has been added in the material and methods section and a new table (table S1) has been provided to show the risk of bias of this articles

Results:

In the tables each referred study should be presented with a proper reference (the same as in the text

We have included the reference in the tables as recommended.

Authors should assess their included studies in any way, e.g. while using Newcastle-Ottawa Scale (NOS)

We have included a ne table (table S1) to show the risk of bias for the articles. As using Newcastle-Ottawa Scale (NOS) is focus on cohort and case-control studies, we employed the National Institutes of Health quality assessment tool for cross-sectional studies

Discussion:

Authors should also indicate other limitations of the study assciated e.g. with the numer of the studies conducted so far which were possible to be included, or subjective character of the data.

 We have included some more limitations regarding the subjective character of the data. The following sentence has been added: Moreover, some the assessment of these symptoms is subjective with may also increase the variability.

Conclusions:

Authors should try to formulate any specific conclusion, associated e.g. with the symptom influecing quality of life in the most signifficant way.

We have included specific conclusion associated with the symptom influencing QoL in a most significant way. The following sentence has been added: Pain might be the symptom most related with impairment in QoL due to its high fre-quency and its subjective component. Malodour is the least studied symptom and could have a major effect on interpersonal relationships.

Authors Contributions:

Authors should present specific actions which were conducted within the study – e.g. what do Authors mean by „validation”, if there is no validation presented?

Thank you for your appreciation. Maybe we understood incorrectly the task. Under Validation of a manuscript, we consider not the task of validation of for example a questionnaire, we consider the validation of the content of the manuscript which is performed by both senior authors, AML and SAS.

It seems that majority of Authors did not present in the manuscript preparation. It is a serious risk of the guest authorship procedure, which is forbidden. They should be either not included as authors (and presented only in Acknowledgements section), or their participation in manuscript preparation should be indicated.

Thank you for inquiry, we also believe that guest authorship needs to be hunted down and eradicated. Clinic investigation is hard, require a lot of effort and time. All the authors of this manuscript are clinical investigator we treat patients during our working hours and time for research comes from our personal and free time. We are totally in line with you.  All researchers included as author contribute to the manuscript preparation. In fact this article include only seven authors, not many for a systematic review and meta-analysis a complicate design that require the collaboration of several people. Author contribution is provided at the end of the manuscript. Author Contributions: Conceptualization, A.M.-L. (Alejandro Molina-Leyva) and T.M.-V.; methodology, A.M.-L. (Alejandro Molina-Leyva) and T.M.-V.; software, J.A.R.P and P.D.-C.; val-idation, A.M.-L. (Alejandro Molina-Leyva) and S.A.-S.; formal analysis, A.M.-L. (Antonio Mar-tinez-Lopez), C.C.-B. and T.M.-V.; investigation, A.M.-L. (Antonio Martinez-Lopez), A.M.-L. (Alejandro Molina-Leyva), C.C.-B. and T.M.-V.; resources, A.M.-L. (Alejandro Molina-Leyva); data curation, J.A.R.P and P.D.-C. and T.M.-V.; writing—original draft preparation, T.M.-V.; writ-ing—review and editing, A.M.-L. (Alejandro Molina-Leyva); visualization, A.M.-L. (Alejandro Molina-Leyva) and S.A.-S.; project administration, A.M.-L. (Alejandro Molina-Leyva); funding acquisition, S.A.-S. All authors have read and agreed to the published version of the manuscript.

Round 2

Reviewer 2 Report

The manuscript entitled „The burden of hidradenitis suppurativa signs and symptoms in quality of life: systematic review and meta-analysis” presents interesting issues but the manuscript should be corrected.

Materials and Methods:

Authors should precisely describe what data were extracted from the studies – it seems that they extracted more data than listed in Materials and Methods Section (e.g. BMI).

Authors should precisely describe applied methodology of meta-analysis

Authors Contributions:

If there was no validation within the manuscript authors should not present it within this section

In order to avoid the risk of misunderstanding, Authors should declare for all of them participation in manuscript preparation.

Author Response

The manuscript entitled „The burden of hidradenitis suppurativa signs and symptoms in quality of life: systematic review and meta-analysis” presents interesting issues but the manuscript should be corrected.

 Thank you for your comments

Materials and Methods:

Authors should precisely describe what data were extracted from the studies – it seems that they extracted more data than listed in Materials and Methods Section (e.g. BMI).

We have now included all the data extracted from the studies. The following sentence has been added: The variables assessed were study design, author, country, level of scientific evi-dence according to the Centre for Evidence-Based Medicine, number of participants, age, sex, BMI (kg/m2), smoking habit, disease duration, disease severity (Hurley stage), HS symptoms and aspects of QoL evaluated, QoL and symptoms assessment tools and scores, correlation between symptoms and QoL.

Authors should precisely describe applied methodology of meta-analysis

 We have tried to explain more precisely the methodology of meta-analysis. The following paragraph has been added in the material and methods section: The mean DLQI and NRS for symptoms was calculated by a random effect me-ta-analysis weighted by the study sample size. To estimate absolute mean effect of DLQI and NRS for each symptom, the mean, standard deviation and sample size were extracted from the studies. Research with unclear or incomplete reporting were ex-cluded from the meta-analysis. To generate valid estimates, studies were weighed ac-cording to their sample-size. Forest plots were constructed to assess the distribution of the data and summarize the effect size and their 95% CIs. Quantifying of Heterogeneity was evaluated using Cochrane Q statistic, an intermediary statistic employed to obtain a more useful measure of heterogeneity, the I2. Assuming a high heterogenicity be-tween studies, we used a random effects model to calculate the outcome. Microsoft Excel version 2016, Redmond, Washington, The USA, was used to run this data[22].

Authors Contributions:

If there was no validation within the manuscript authors should not present it within this section

Following your recommendations, we have omitted the validation in the author contribution.

In order to avoid the risk of misunderstanding, Authors should declare for all of them participation in manuscript preparation.

Thank you for the comment. We have added the following sentence in the Author contribution section: All authors declare have been participated in manuscript preparation.